# Genetic Characteristics of the Rat Fibroblast Cell Line Rat-1

**DOI:** 10.3390/cells14010021

**Published:** 2024-12-29

**Authors:** Thomas Liehr, Stefanie Kankel, Eva Miriam Buhl, Sarah K. Schröder-Lange, Ralf Weiskirchen

**Affiliations:** 1Institute of Human Genetics, Jena University Hospital, Friedrich Schiller University, D-07747 Jena, Germany; stefanie.kankel@med.uni-jena.de; 2Electron Microscopy Facility, Institute of Pathology, RWTH University Hospital Aachen, D-52074 Aachen, Germany; ebuhl@ukaachen.de; 3Institute of Molecular Pathobiochemistry, Experimental Gene Therapy and Clinical Chemistry (IFMPEGKC), RWTH, University Hospital Aachen, D-52074 Aachen, Germany; saschroeder@ukaachen.de

**Keywords:** fibroblasts, rat, in vitro model, SKY analysis, STR profiling, karyogram, next-generation sequencing, cell authentication

## Abstract

The Rat-1 cell line was established as a subclone of the parental rat fibroblastoid line F2408, derived from Fisher 344 rat embryos. Rat-1 cells are widely used in various research fields, especially in cancer biology, to study the effects of oncogenes on cell proliferation. They are also crucial for investigating signal transduction pathways and play a key role in drug testing and pharmacological studies due to their rapid proliferation. Therefore, Rat-1 cells are an indispensable research tool. While some cytogenetic information on their basic chromosomal features is available, detailed genomic analyses, such as karyotype analysis, short tandem repeat (STR) profiling, and whole-genome sequencing, have not been thoroughly conducted. As a result, the genetic stability and potential variations in Rat-1 cells over extended culture periods are poorly understood. This lack of comprehensive genetic characterization can limit the interpretation of experimental results and requires caution when generalizing findings from studies using this cell line. In this study, we describe the genetic characterization of the Rat-1 cell line. We established a karyotype, performed multicolor fluorescence in situ hybridization (mFISH), identified chromosomal losses and gains, and defined an STR profile for Rat-1 with 31 species-specific markers. Interestingly, the chromosomal imbalances found in Rat-1 cells resemble those found in human epithelioid sarcoma or liposarcoma. Additionally, we analyzed the transcriptome of Rat-1 cells through mRNA sequencing (mRNA-Seq) using next-generation sequencing (NGS). Finally, typical features of these fibroblastic cells were determined using electron microscopy, Western blotting, and fluorescent phalloidin conjugates.

## 1. Introduction

Well-characterized cell lines are essential tools in biomedical research, providing a controlled environment for studying cellular processes, disease mechanisms, and therapeutic interventions [1]. The Rat-1 cell line was originally derived from a subclone of the parental F2408 line from Fisher 344 rat embryos [2,3,4]. They exhibit rapid proliferation and anchorage-dependent growth characteristics that make them suitable for diverse experimental applications [2,3,4]. Since their establishment, Rat-1 cells have gained prominence across various fields and have been used extensively in studies investigating aspects of cancer biology, cell cycle, pharmacology, oncogenic transformation, and gene editing, e.g., [5,6,7,8,9]. Particularly, Rat-1 cells have been attractive in experiments investigating cellular transformation because of their low level of spontaneous focus formation [10].

Despite their widespread use, comprehensive genetic characterization of Rat-1 cells remains limited. While very sparse information regarding their karyotype and chromosomal features exists [10], detailed genomic analyses, including detailed karyotype assessment, short tandem repeat (STR) profiling, and whole-genome sequencing, have not been systematically conducted. This gap in knowledge raises important questions about the genetic stability and potential variations within Rat-1 cells over extended culture periods.

Numerous studies have utilized Rat-1 cells to investigate critical biological questions. For instance, research has shown how oncogenes such as *Ras* influence cell proliferation and transformation in cancer models [11,12]. Similarly, these cells have been used to unravel fundamental aspects of cyclin activity [13]. Additionally, Rat-1 cells have been employed to explore signal transduction pathways, like the MAPK/ERK pathway, shedding light on cellular responses to growth factors [14]. They have also played a crucial role in drug testing; studies assessing the efficacy of novel compounds often utilize this cell line due to its rapid growth dynamics [15,16,17].

In this study, we aimed to provide a thorough genetic characterization of Rat-1 cells. We established a karyotype and performed multicolor fluorescence in situ hybridization (mFISH) to elucidate chromosomal composition. Additionally, we defined an STR profile using 31 species-specific markers to confirm the identity and stability of the cell line. Furthermore, we analyzed the transcriptome through mRNA sequencing (mRNA-Seq) via next-generation sequencing (NGS) to gain insights into gene expression patterns. Finally, typical morphological features of these fibroblastic cells were assessed using electron microscopy, Western blotting, and phalloidin staining. By addressing these aspects of genetic characterization, our study provides critical insights that can inform future research utilizing this cell line.

## 2. Materials and Methods

### 2.1. Literature Search

Papers involving Rat-1 cells were found by searching the PubMed database [18] using the search term (“Rat-1 cells” or “Rat-1 fibroblasts” or “Rat1 cells” or “Rat1 fibroblasts”).

### 2.2. Cell Culture

The adherent, immortalized, epithelial-like Rat-1 cell line (CVCL_0492) was initially established and characterized over fifty years ago [2,3,4]. These cells are accessible from the Riken BioResource Research Center (#RCB1830, Tsukuba, Japan). The experiments in this study were conducted at passage numbers pX + 4 to pX + 8, where X denotes the previous unknown passage numbers and N signifies the current passage number in our laboratory. The cells were regularly grown in 10 cm^2^ Petri dishes and maintained in Dulbecco’s Modified Eagle’s Medium (DMEM) supplemented with a 10% fetal bovine serum (FBS), 2 mM L-Glutamine, 1 mM sodium pyruvate, and 1× Penicillin/Streptomycin. The medium was changed every other day, and cells were subcultured at a split ratio of 1:8 to 1:10 twice per week using the Accutase solution (#A6964, Sigma-Aldrich, Taufkirchen, Germany).

### 2.3. Short Tandem Repeat (STR) Profiling

The STR profiling and testing for interspecies contamination in Rat-1 cells were conducted through the cell line authentication service provided by IDEXX (Kornwestheim, Germany) using the CellCheck^TM^ Rat system. This system is a dinucleotide repeat assay that creates a genetic profile of the cells by employing 31 specific STR markers unique to different species.

### 2.4. Preparation of Rat-1 Metaphase Chromosomes, Karyotyping, and Molecular Cytogenetics

Chromosomes from Rat-1 cells were prepared using a standard protocol for metaphase preparation in rat fibroblast cultures, with some minor adjustments [19]. In summary, Rat-1 cells were incubated at 37 °C in T25 flasks until they reached a semi-confluent density. Following treatment with a KaryoMAX colcemid solution (#15212012, Gibco, ThermoFisher Scientific, Schwerte, Germany), the cells were detached using mild trypsin-EDTA (#T4174, Sigma-Aldrich, Merck KGaA, Darmstadt, Germany) and collected into a centrifuge tube. After a short centrifugation step, the cells underwent hypotonic treatment with 0.56% KCl for 30 min at 37 °C before being fixed in a mixture of acetic acid and methanol (1:3). Air-dried chromosome spreads were then created from the fixed cell suspension and used in multicolor fluorescence in situ hybridization (mFISH), as previously described [20]. The commercially available rat probe set 22xRat (MetaSystems, Altlussheim, Germany) was used to detect interchromosomal rearrangements in the chromosomes of Rat-1. Due to the specific setup of this probe set, chromosomes 13 and 14 cannot be distinguished after FISH [21]. For evaluation, 30 metaphases were analyzed using Zeiss Axioplan microscopy equipped with ISIS software (vers. 6.1.1; MetaSystems, Altlussheim, Germany). To establish chromosome banding, metaphases, sorted according to FISH-signals and counterstained by 4′,6-diamidino-2-phenylindole (DAPI), were transformed into an ‘inverted DAPI-banding’ pattern, as possible with one click in the applied software.

### 2.5. Virtual Comparative Genomic Hybridization

Using the data obtained from mFISH in combination with inverted DAPI-banding, approximate losses and gains of chromosomal regions in rat chromosomes were identified. These regions were then mapped to their approximate molecular positions (based on Rat RGSC 5.0/rn5) using the UCSC browser. By utilizing the “View, In Other Genomes” function, these regions were translated to the human genome (build: GRCh37/hg19) and homologous regions with gains or losses determined. These copy number alterations (CNAs) were then compared to spindle cell-shaped tumors of the skin, such as epithelioid sarcoma [22] and liposarcoma [23].

### 2.6. Next-Generation Sequencing and Data Analysis

High-quality RNA was extracted from five 100 mm^2^ plates of Rat-1 cells grown under basal conditions to 80% confluence using a well-established method involving CsCl_2_ density gradient centrifugation. Therefore, confluent cells were lysed and homogenized in a buffer containing guanidine thiocyanate. The resulting solution was then layered onto a cesium chloride cushion and centrifuged for 21 h at 21 °C and 25,000 rpm in a Beckman SW41 rotor. After centrifugation, the pellet was resuspended in sterile water, purified through ethanol precipitation, and finally dissolved in sterile water. RNA concentration, purity, and quality were assessed via UV spectroscopy and on the Agilent 4200 TapeStation automated platform (Agilent Technologies Inc., Waldbronn, Germany). Following this, ribosomal RNA was depleted, and mRNA was converted into a library of template molecules suitable for cluster generation and DNA sequencing using the NEBNext Multiplex Oligos for Illumina Index Primers Set 1 kit, which includes preformed adaptors and primers. Sequencing was performed on the Illumina platform (Illumina Inc., San Diego, CA, USA) using pre-filled, ready-to-use MiSeq Reagent kit V2, 300-cycles cartridges (Illumina Inc.). All sequencing results were subsequently transformed into fastq data files. The construction and sequencing of the cDNA library took place at the IZKF Genomic Facility of the University Hospital Aachen.

Before any downstream analysis could be performed, FASTQ files were generated using bcl2fastq (Illumina). To facilitate reproducible analysis, samples were processed using the publicly available nf-core/RNA-seq pipeline version 3.12 [24] implemented in Nextflow 23.10.0 [25]. In brief, lane-level reads were trimmed using Trim Galore 0.6.7 [26] and aligned to the human genome (GRCh39) using STAR 2.7.9a [27] with the *Rattus norvegicus* reference genome (build: mRatBN7.2). Gene-level and transcript-level quantification was performed using Salmon v1.10.1 [28]. The results were exported as a length-normalized Transcripts Per Million (TPM) quantification of transcript abundances.

### 2.7. Electron Microscopic Cell Analysis

Electron microscopic analysis was conducted following established protocols [29]. In summary, cells were fixed in 1× phosphate-buffered saline (PBS) with 3% glutaraldehyde, rinsed with 0.1 M Soerensen’s phosphate buffer, and post-fixed in 1% osmium tetroxide. Dehydration was carried out using a series of ethanol solutions (30% to 100%). Subsequently, specimens were incubated in propylene oxide, followed by Epon resin mixtures, and polymerized at 90 °C for two hours. Ultrathin sections (90–100 nm) were cut and placed on Cu/Rh grids, stained with uranyl acetate and lead citrate, and observed using a Zeiss Leo 906 transmission electron microscope at 60 kV. Images were captured at magnifications indicated in the respective figure legends.

### 2.8. Western Blot Analysis

Protein extracts, quantification, and Western blot analysis were conducted following established protocols. In brief, equal amounts of protein extracts (40 µg/lane) were heated at 80 °C for 10 min and separated using 4–12% Bis-Tris gels (Invitrogen, Thermo Fisher Scientific, Schwerte, Germany) under reducing conditions with MES running buffer (50 mM (2-(N-morpholino)ethanesulfonic acid), 50 mM Tris-base, 0.1% Sodium dodecyl sulfate, 1 mM Ethylenediaminetetraacetic acid, pH 7.3). The proteins were electro-blotted onto a 0.45 µm nitrocellulose membrane (#GE10600002, Amersham^TM^ Protran^®^ Western-Blotting Membranes, Merck, Darmstadt, Germany), with transfer efficiency confirmed by Ponceau S staining. Blocking was performed in Tris-buffered saline (50 mM Tris, 150 mM NaCl) containing 0.1% Tween 20 with 5% non-fat milk powder. The membranes were probed with specific antibodies for collagen type I, ferritin heavy chain (Fth1), fibronectin, vimentin, and glyceraldehyde 3-phosphate dehydrogenase (GAPDH). The detection of primary antibodies was achieved using HRP-conjugated secondary antibodies and the Supersignal™ chemiluminescent substrate (Perbio Science GmbH, Bonn, Germany). Details about all antibodies used are provided in Table 1.

### 2.9. Phalloidin Stain

Microfilament staining was performed as previously outlined [30]. In brief, 30,000 Rat-1 cells were seeded on glass coverslips in a 24-well plate. After 48 h, the medium was removed, and cells were washed with PBS before being fixed in 3.7% paraformaldehyde (buffered with phosphoric acid to pH 7.4) for 20 min. Following fixation, the cells were permeabilized with a precooled solution of 0.1% sodium citrate and 0.1% Triton X-100 for 3 min on ice. After additional PBS washes, nonspecific binding sites were blocked with PBS containing 50% FBS and 0.5% bovine serum albumin for one hour. Next, the cells were stained in the dark with either a 1× diluted Rhodamine-Phalloidin solution (R415, Thermo Fisher Scientific, Schwerte, Germany) or an Alexa Fluor^TM^ 488 Phalloidin (#A12379, Thermo Fisher Scientific) for 20 min. Subsequently, the nuclei were counterstained with 4′,6-diamidino-2-phenylindole (DAPI, #D1306, Thermo Fisher Scientific) for 15 min in the dark. Finally, the samples were mounted with the PermaFluor^TM^ aqueous mounting medium (#TA-030-FM, Thermo Fisher Scientific) and analyzed under a Nikon Eclipse E80i fluorescence microscope with NIS-Elements Vis software (vers. 3.22.01). For a detailed hands-on protocol with instructions, refer to [30].

## 3. Results

### 3.1. Usage of Rat-1 Cells in Biomedical Research

A search on PubMed conducted on 19 November 2024 using the terms (“Rat-1 cells” or “Rat-1 fibroblasts” or “Rat1 cells” or “Rat1 fibroblasts”) yielded a total of 987 papers, highlighting the extensive research and interest surrounding Rat-1 cells and their applications in various scientific studies. In the past, Rat-1 cells have been utilized to analyze the effects of oncogenes and viruses on cellular behavior. Specifically, researchers have employed this cell line to study how specific oncogenes influence cell proliferation, transformation, and tumorigenesis. Additionally, Rat-1 cells have served as a model for investigating viral interactions with host cells, providing insights into viral pathogenesis and the mechanisms of infection. In contrast, Rat-1 cells are currently most frequently used in studies focused on investigating drug effects and conducting transcriptional research, suggesting that their characteristics make them a valuable model for exploring various biological processes and testing therapeutic interventions.

### 3.2. Phenotypic Appearance of Rat-1 Cells

Rat-1 cells in culture typically display a distinct phenotypic appearance. They are usually elongated and spindle-shaped, with a prominent nucleus and well-developed cytoplasm (Figure 1). In monolayer cultures, Rat-1 cells tend to grow in a characteristic “haphazard” or “random” arrangement, often forming overlapping layers. When cultured at low confluence, the cells can extend long processes or filopodia, which contribute to their ability to migrate and interact with the extracellular matrix. Similarly to other fibroblastic cell lines, Rat-1 cells exhibit a high degree of plasticity, enabling them to adapt their morphology based on the surrounding environment and culture conditions.

### 3.3. Electron Microscopic Analysis of Rat-1 Cells

Electron microscopic analysis is a powerful technique used to investigate the fine structure of cells at a nanometer scale, revealing intricate details of cellular components and their organization. This level of detail is crucial for understanding cell morphology, as it enables the visualization of organelles, membranes, and other subcellular structures that are often not discernible with traditional light microscopy. In addition to the elongated spindle-shaped cell morphology, Rat-1 cells have a slightly elongated, centrally located nucleus, which accounts for about 25–50% of the cell volume. The chromatin is loosened, indicating high metabolic activity of the cells. This is a characteristic of cell lines of fibroblastic origin (Figure 2A,B). The nuclei of Rat-1 cells display one or more nucleoli, which are prominent and well-defined structures within the nucleus. The organelles are distributed evenly within the cytoplasm. The mitochondria appear as elongated or spherical organelles of different sizes with a double membrane structure, featuring intricate inner folds known as cristae that increase surface area for the respiratory chain (Figure 2C–E). The cytoplasm of Rat-1cells is enriched by small vesicles and can contain electron-dense bodies and vacuoles. The endoplasmic reticulum of Rat-1 cells appears as a network of membranous tubules and flattened sacs, with a rough endoplasmic reticulum characterized by ribosomes on its cytoplasmic surface (Figure 2F).

### 3.4. Expression of Typical Fibroblast Markers in Rat-1 Cells

#### 3.4.1. Next-Generation Sequencing

Fibroblasts typically express a variety of genes that are essential for maintaining the extracellular matrix and supporting tissue structure. Key genes include those that encode collagen types I, III, and V, which provide tensile strength and structural support to connective tissues. Transcriptional profiling using next-generation sequencing (NGS) revealed that Rat-1 cells express high levels of various collagens (Table 2).

Additionally, fibroblasts express fibronectin and laminin, which facilitate cell adhesion and migration. Other important genes include those involved in the synthesis of proteoglycans and glycoproteins that contribute to the integrity of the extracellular matrix. Furthermore, fibroblasts may also express growth factors, such as transforming growth factor-β (TGF-β) and platelet-derived growth factor (PDGF), playing significant roles in wound healing and tissue repair processes. All of these genes are expressed in Rat-1 cells, supporting the idea that Rat-1 cells are of fibroblastic origin (Table 3).

In our NGS analysis, we identified a total of 27,424 different transcripts (Appendix A). The highest expression was found for the eukaryotic translation elongation factor 1 alpha1 (*Eef1a1*), which was expressed at 12,161.3729 TPM. This is surprising as this gene located in 8q31 is only present in one copy in the Rat-1 genome, according to karyotype analyses (see below) The lowest expression was observed for the coiled coil domain containing 88 (*Ccdc88a*) with an expression level of 0.000031 TPM; it is located on 14q22 and present in 2 copies. Additionally, our analysis revealed that Rat-1 cells express fibroblast-specific protein 1 (*Fsp1*) also known as S100 calcium-binding protein A4 (*S100a4*), integrin subunit beta 1 (*Itgb1*), Theta antigen (*Thy1*/CD90), and the platelet-derived growth factor receptor alpha (*Pdgfra*) and beta (*Pdgfrb*). The expression of the extracellular matrix glycoprotein Tenascin-C (*Tnc*) was rarely expressed (0.154453 TPM), underpinning the notion that the immortalized fibroblast line Rat-1 has non-metastatic properties [10,31,32]. Additionally, our data confirmed earlier findings that genes encoding the period circadian regulator 1 (*Per1*), *Per2*, nuclear receptor subfamily 1, group D, member 1 (*Nr1d1*), D-box-binding PAR bZIP transcription factor (*Dbp*), and thyrotroph embryonic factor (*Tef*) are expressed in Rat-1 cells (Table 4), which were previously associated with circadian gene expression in mammalian tissue and Rat-1 cells [33].

All these findings are in agreement that Rat-1 cells are of fibroblastic origin and show that our expression list provides a good reference list of what is expressed in Rat-1 cells.

#### 3.4.2. Analysis of Protein Expression and Cytoskeletal Organization in Rat-1 Cells

To validate our NGS data, we conducted Western blot analysis on key markers indicative of the fibroblastic phenotype. Our results aligned with the NGS data, demonstrating the expression of collagen type I, fibronectin, and vimentin in Rat-1 cells. These genes play a role in extracellular matrix formation and cell adhesion (Figure 3). Additionally, we observed the presence of ferritin, a protein responsible for iron storage. However, the expression of ferritin in Rat-1 cells was lower when compared to the hepatic stellate cell line PAV-1, which was used as a control cell line and known to express high levels of ferritin [34].

The cytoskeleton is a crucial component of fibroblasts, playing a significant role in maintaining cell elasticity in response to environmental stimuli and enabling the cells to withstand forces that exceed those required for maintaining their shape and movement [35]. Consistent with the suggested fibroblastic characteristics of Rat-1 cells, and in accordance with previous findings [5], we observed that these cells develop a strong network of cytoplasmic microfilaments, as evidenced by staining with phalloidin conjugates (Figure 4).

### 3.5. Karyotype Based on Molecular Cytogenetic Analyses

We conducted mFISH and simultaneously established a conventional karyogram using inverted DAPI-banding on the Rat-1 cell line. This was performed to identify any chromosomal abnormalities and gain a more comprehensive understanding of the chromosomal landscape (Figure 5).

We found that Rat-1 cells have a karyotype of a rat with between 37 and 43 chromosomes (2n) and an XY sex chromosome configuration, suggesting that this rat is of male origin. It features a derivative chromosome 2 formed by a translocation between chromosome 2 and chromosome 19, with breakpoints located on chromosome 2q34 and chromosome 19p11. Additionally, there is the absence of one copy of chromosome 3 and another derivative chromosome involving a translocation between chromosomes 3 and 4, with approximate breakpoints at 3q22 and 4p11. Furthermore, there is also a monosomy of chromosomes 4 and 8 and a trisomy of chromosome 7. Complex rearrangements are noted in derivative chromosome 12, which includes segments from other chromosomes inserted into its structure. Lastly, another derivative form of chromosome 19 shows further complex rearrangements involving segments from other chromosomes as well. Overall, this karyotype indicates significant chromosomal rearrangements that can have various implications for genetic studies or understanding disease mechanisms in rat models.

### 3.6. Virtual Comparative Genomic Hybridization

Next, we translated the results obtained from mFISH and karyotype analyses into gains and losses to emphasize copy number variations in the Rat-1 genome (Table 5).

The losses and gains from Table 3 are projected in the human genome in Figure 6. It is interesting that almost all affected regions showed a loss or gain of one copy, while regions 19pter->19p11 and 2q?2->2q?2 were present in 2 and 4 additional copies, respectively. It remains to be determined which tumor suppressor genes are located in regions with loss of copy numbers and which of the oncogenes present in amplified regions are responsible for the immortalization of Rat-1 cells.

Interestingly, 17 out of 40 (42.5%) CNAs found in Rat-1 were present in epithelial sarcoma, and 15 out of 40 (37.5%) were present in liposarcoma (Table 6). Since these tumors have the potential for immortality, this suggests that these CNA patterns may be prerequisites for the immortalization of this cell type.

### 3.7. Short Tandem Repeat Analysis

Short tandem repeat (STR) profiling is essential for verifying the identity and authenticity of cell lines in culture to identify contamination or misidentification. In previous studies, we identified several genetic characteristics and STR profiles for the rat cell lines HSC-T6 [36], CFSC-2G [29], and PAV-1 [34] using a panel of 31 variable repeat sequences. Using the same panel of markers, we now have conducted STR profiling for the Rat-1 cell line. The obtained STR profile indicates that the Rat-1 cell line possesses a distinct STR profile that is markedly different from other rat fibroblastic cells that we have profiled before (Table 7; Appendix A).

The obtained STR profile is crucial for future authentication, as it provides a reliable reference for verifying the identity and integrity of the cells in subsequent studies. This allows for the identification of misidentification and contamination, ensuring the validity of experimental results when using Rat-1 as an experimental tool.

## 4. Discussion

The Rat-1 fibroblast cell line has been a crucial tool in various fields of biomedical research, particularly in cancer biology and pharmacology. It is important to note that Rat-1 cells were often chosen due to their reputation as a genetically unaltered fibroblast cell line. However, despite its extensive use, the genetic characteristics of Rat-1 cells have not been thoroughly examined until now. This study aims to provide a comprehensive understanding of the genetic stability and identity of Rat-1 cells through molecular cytogenetic-based karyotype analysis, short tandem repeat (STR) profiling, mRNA sequencing, and morphological assessments.

Our karyotype analysis revealed significant chromosomal rearrangements within Rat-1 cells. The presence of a variable number of chromosomes, ranging from 37 to 43, along with complex rearrangements, suggests a level of genetic instability that could impact experimental outcomes. This instability may lead to variations in gene expression and cellular behavior over prolonged culture periods, which researchers must consider when interpreting results from experiments involving this cell line. Similar findings have been observed in other immortalized or primary cell lines where chromosomal abnormalities or instability have been linked to altered phenotypic characteristics [37,38,39,40]. Therefore, our results emphasize the necessity of regular genetic characterization to maintain the reliability of experimental models.

It is worth noting that the first cytogenetic analysis in Rat-1 cells was reported by Reynolds and colleagues nearly four decades ago, revealing a prominent acrocentric chromosome not typically found in normal rat cells [10]. This could possibly refer to the +der(19)(19qter->19p11::2q?2->2q?2::6q11->6qter) found in the present analysis. The previous report [10] also sporadically identified small chromosomal fragments (minutes) and a significant long acrocentric chromosome. The latter likely refers to the der(12)(12pter->12q1?2::2q?2->2q?2::2q?2->2q?2::19p11->19pter), while no hint of chromosomal fragments was found in this study. Future studies should explore gene expression patterns and functional implications of both chromosomal rearrangements and CNAs. In-depth analyses of losses and gains could be extended to aid in identifying cell-specific genetic stability, oncogenic processes, and the relevance of Rat-1 cells in cancer research or drug development. The relatively large agreement of CAN patterns observed in Rat-1 and human sarcomas; however, suggests that Rat-1 is more of a model for human sarcomas than suited as a substitute for normal rat fibroblasts.

The STR profiling conducted in this study further supports the uniqueness of the Rat-1 cell line by establishing a distinct genetic identity compared to other rat fibroblastic lines. This aspect is crucial for ensuring authenticity and preventing cross-contamination in laboratory settings. The establishment of an STR profile serves as an essential tool for future studies involving Rat-1 cells and reinforces best practices in cell line authentication. The determined STR profile can also provide insights into genetic stability over time, allowing researchers to assess changes that may occur during prolonged culturing.

In addition to chromosomal analyses, we performed transcriptomic profiling using next-generation sequencing (NGS). Our findings indicate that Rat-1 cells express key fibroblast markers, such as collagen types I and III, fibronectin, vimentin, integrins, and several laminin subunits. These markers are essential for maintaining extracellular matrix scaffold organization and integrity, and facilitating cellular adhesion and migration, characteristics intrinsic to fibroblasts [41,42,43,44]. The high expression levels observed align with previous literature, indicating that fibroblasts play critical roles in wound healing and tissue repair processes [45,46]. Moreover, our data corroborate earlier studies suggesting that Rat-1 cells maintain their fibroblastic phenotype even after prolonged culture periods [2,3,4]. Our NGS data set also demonstrates that Rat-1 cells express PDGF, FGF, TGF-β, vascular endothelial growth factor (VEGF), fibroblast growth factor (FGF) family, and many other mediators that can act as autocrine or paracrine factors (Appendix A), suggesting Rat-1 cells as a simple and suitable model for addressing aspects of wound healing.

Morphological assessments using electron microscopy revealed distinct features consistent with fibroblast-like morphology, including elongated spindle shapes and well-defined organelles such as mitochondria and the endoplasmic reticulum. These observations further support our transcriptomic findings, which highlight the maintenance of typical fibroblast characteristics within this cell line. The cytoskeleton in fibroblasts is a dynamic and complex network of protein filaments that provides structural support, shape, and mechanical stability to the cells [47]. It consists primarily of three main types of filaments: microfilaments (actin filaments), intermediate filaments, and microtubules. Microfilaments are involved in cell motility and shape changes, playing a crucial role in processes such as wound healing. Intermediate filaments provide tensile strength and help maintain the integrity of the cell under stress. Microtubules facilitate intracellular transport and play a significant role in cell division. Together, these components enable fibroblasts, such as Rat-1 cells, to perform their essential functions in tissue repair and maintenance effectively. Furthermore, the finding that crucial mediators belonging to the family of cytoskeleton-associated proteins are expressed highlights the suitability of Rat-1 cells for respective studies.

The experiments in our study were conducted at a low passage number and should be highly reproducible at other locations when also conducted at a low passage number. However, we acknowledge that cells considered “identical” may actually differ significantly. The scientific community has learned from HeLa cells, where various studies have demonstrated that HeLa cells from different sources or laboratories exhibit a high degree of variability in genomic instability, gene expression, and protein profiles [48,49]. Furthermore, single-cell DNA and RNA sequencing with HeLa-CCL2 cells have revealed significant transcriptomic heterogeneity that correlates with copy number variations in the HeLa-CCL2 genome [50]. This underscores the notion that biological features of cell lines can change over time and across different environments, emphasizing the need for careful characterization of any cell line used in research.

It is noteworthy that our study provides valuable insights into the genetic stability and biological characteristics of Rat-1 cells. However, it also raises important questions about how these factors may influence experimental outcomes in various research applications. For example, potential variations in gene expression due to chromosomal instability could impact drug response assays or oncogene transformation studies conducted using this model. Therefore, researchers should exercise caution when generalizing findings from experiments involving Rat-1 cells without considering their unique genetic background.

## 5. Limitations of This Study

Our study on the genetic characteristics of the Rat-1 fibroblast cell line has several limitations that should be acknowledged. Firstly, despite providing a comprehensive analysis, there still may be uncharacterized genetic variations or mutations within the Rat-1 cell line that were not detected in this study. This could limit our understanding of its full genetic profile and stability. Additionally, the presence of significant chromosomal rearrangements and instability over extended culture periods raises concerns about the reproducibility of experimental results in other laboratories using different passage numbers. Variations in gene expression due to genomic instability could affect findings derived from this cell line.

Moreover, while the chromosomal imbalances found in Rat-1 cells have been compared to human sarcomas in our study, it is crucial to consider interspecies differences when extrapolating results to human conditions. The degree to which findings from Rat-1 cells can accurately represent human disease mechanisms remains uncertain. Therefore, Rat-1 cells might not be an appropriate model for studying aspects of sarcomas. The specific culture conditions used in the study, including passage numbers and media composition, may also influence cellular behavior and gene expression patterns, potentially affecting the generalizability of results across different laboratories or studies.

Furthermore, although the study primarily focuses on certain genetic markers and phenotypic characteristics associated with fibroblasts, it might overlook other relevant biological processes or pathways that could provide further insights into the functionality of Rat-1 cells.

While transcriptomic and proteomic analyses were performed in this study, functional assays, such as serum stimulation experiments, validating how these genetic characteristics translate into biological behavior, such as proliferation rates or response to drugs, were not extensively covered. Lastly, if comparisons of NGS data were made with other non-fibrogenic cell lines or tumor-derived cells, this would have helped to identify Rat-1 specific markers and strengthen the conclusions regarding similarities and differences among them. Additionally, the study does not account for temporal changes in gene expression or chromosomal stability that may occur over time during prolonged culture periods, which could significantly impact experimental outcomes.

Overall, while this study provides valuable insights into the Rat-1 cell line’s genetic characteristics, these limitations highlight areas for further research and caution when interpreting results derived from this model system.

## 6. Conclusions

In conclusion, our investigation into the genetic characteristics of the Rat-1 cell line highlights its potential utility as a reliable model system for human sarcoma rather than for normal rat fibroblasts. It also draws attention to inherent limitations associated with genomic variability. Future studies should aim to continuously monitor genetic stability within established cell lines like Rat-1 to enhance reproducibility across scientific investigations. We hope that this foundational work will pave the way for a more nuanced understanding and application of Rat-1 cells in diverse areas of biomedical research.

## Figures and Tables

**Figure 1 cells-14-00021-f001:**
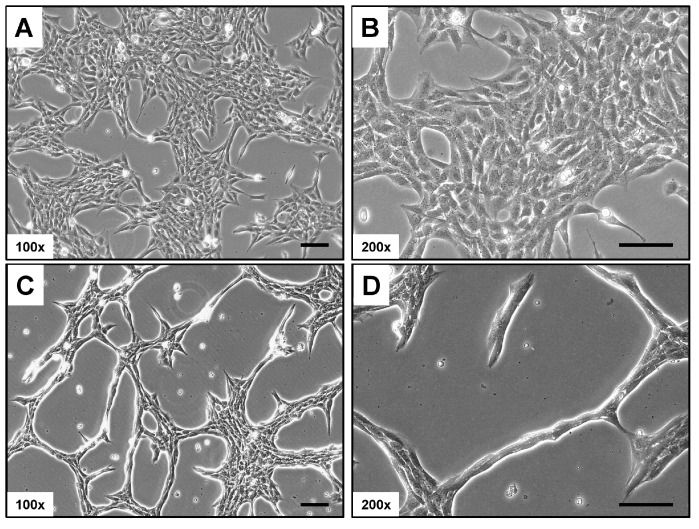
Light microscopic appearance of Rat-1 cells. (**A**,**B**) Rat-1 cells in culture are elongated and spindle-shaped with a prominent nucleus. They typically grow in a “haphazard” arrangement, forming overlapping layers. (**C**,**D**) At low confluence, they extend filopodia, enhancing migration and interaction with the extracellular matrix, and display high plasticity in response to their environment. Magnifications are 100× (**A**,**C**) and 200× (**B**,**D**). Scale bars represent 100 µm.

**Figure 2 cells-14-00021-f002:**
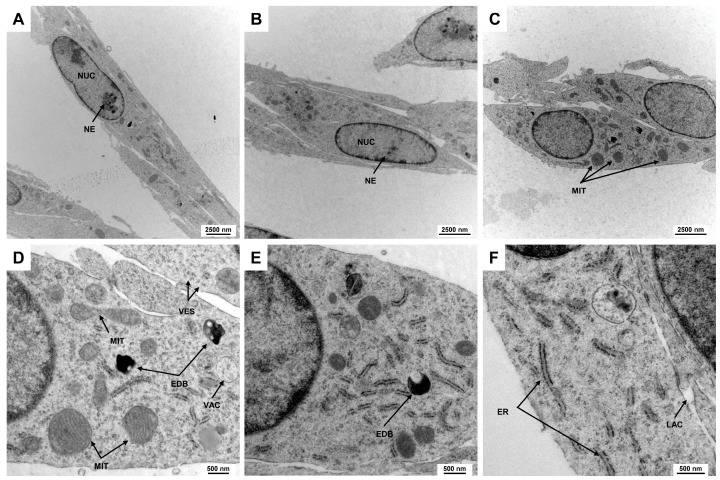
Electron microscopic appearance of Rat-1 cells. (**A**,**B**) Rat-1 cells exhibit an elongated spindle-shaped morphology with a centrally located nucleus (NUC) occupying 25–50% of the cell volume. The nuclei contain prominent nucleoli (NE). (**C**) Mitochondria (MIT) appear as elongated or spherical organelles with double membranes and cristae that increase metabolic surface area. (**D**,**E**) The cytoplasm is filled with small vesicles (VES), electron-dense bodies (EDBs), and vacuoles (VACs). (**F**) The endoplasmic reticulum of Rat-1 cells forms a network of membranous tubules and flattened sacs, with rough endoplasmic reticulum identifiable by ribosomes on its surface. The images were taken at (**A**) 3597×, (**B**,**C**) 4646×, (**D**,**E**) 16,700×, and (**F**) 21,560× magnification, respectively.

**Figure 3 cells-14-00021-f003:**
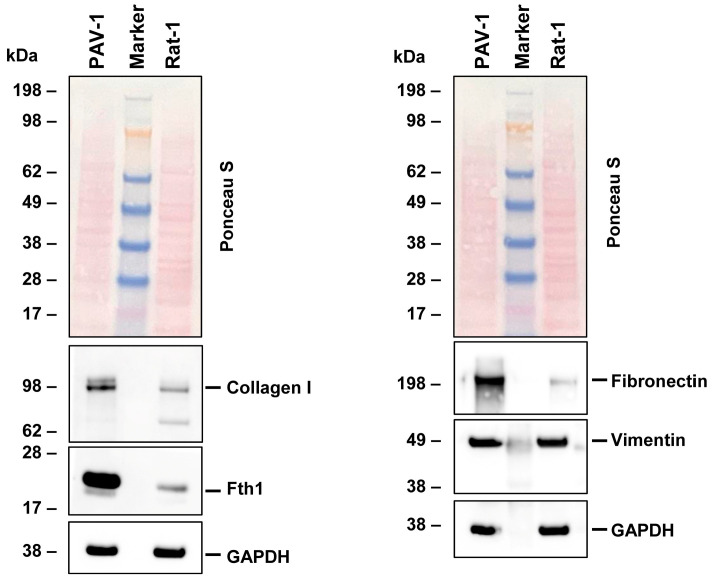
Protein expression of characteristic fibroblast markers in Rat-1 cells. Cell protein extracts were prepared from PAV-1 and Rat-1 cells, then analyzed by Western blot analysis (40 µg protein/lane) for the expression of collagen type I, ferritin heavy chain (Fth1), fibronectin, and vimentin. Ponceau S staining and probing with a glyceraldehyde 3-phosphate dehydrogenase (GAPDH)-specific antibody served as controls to ensure equal protein loading.

**Figure 4 cells-14-00021-f004:**
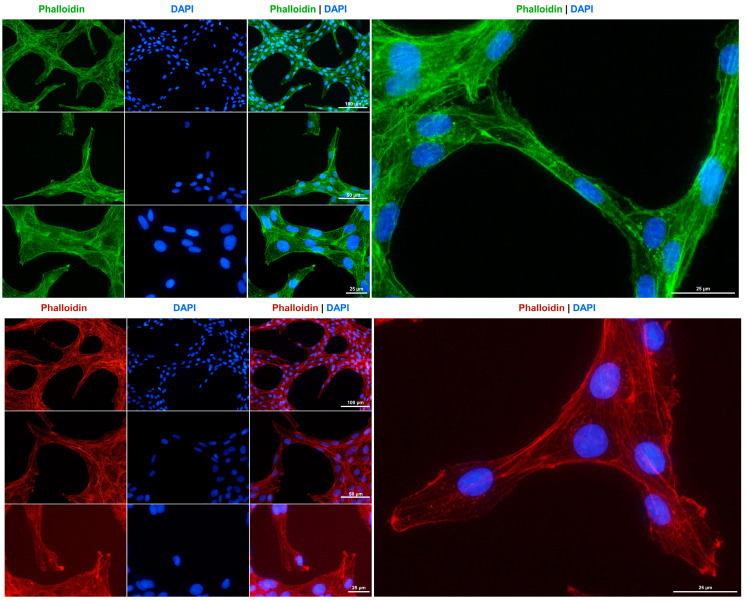
F-actin cytoskeleton staining in Rat-1 cells. The cytoskeleton of cultured Rat-1 cells was labeled with either an Alexa Fluor 488^TM^-labeled Phalloidin (green) or a Phalloidin Rhodamine conjugate (red). Nuclei were counterstained with DAPI (blue). Images were captured using a Nikon Eclipse E80i fluorescence microscope at 200×, 400×, or 600× magnification. Scale bars at the different magnifications are provided.

**Figure 5 cells-14-00021-f005:**
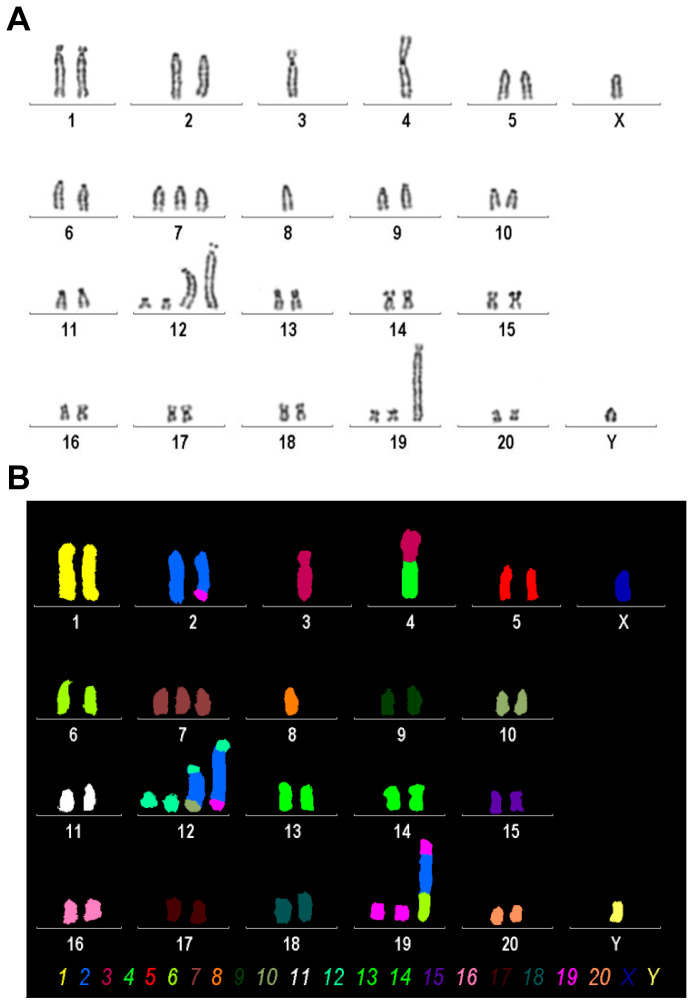
Karyogram analysis and mFISH results of the Rat-1 cell line. (**A**) A representative image of inverted DAPI-banding is shown. (**B**) mFISH result of the same metaphase, as shown in A, using the commercially available 22xRat multicolor FISH probe; interchromosomal rearrangements in Rat-1 chromosomes are visible as color changes within single chromosomes; the color code for each chromosome is provided below. Evaluation of this analysis revealed the following karyotype: 37~43<2n>,XY,der(2)t(2;19)(q34;p11),-3,der(4)t(3;4)(q?22;p11),-4,+7,-8,+der(12)(12pter->12q1?2::2q?2->2q?2::10q?3->10qter),+der(12)(12pter->12q1?2::2q?2->2q?2::2q?2->2q?2::19p11->19pter), +der(19)(19qter->19p11::2q?2->2q?2::6q11->6qter)[cp30].

**Figure 6 cells-14-00021-f006:**
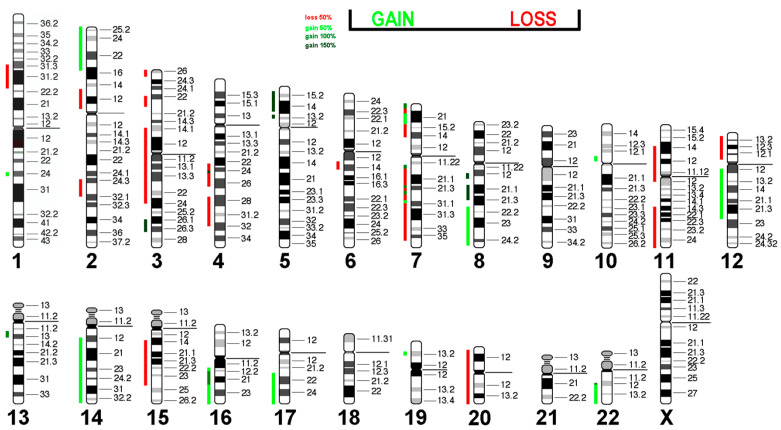
Virtual Comparative Genomic Hybridization results for the Rat-1 cell line were translated into human genome. Copy number alterations are depicted in a color code. Shades of red indicate losses, while green indicate gains.

**Table 1 cells-14-00021-t001:** Primary and secondary antibodies used for Western blot analysis ^1^.

Antibody	Cat. No.	Company	Dilution	Clonality
Collagen I	14695-1-AP	Proteintech	1:1000	r pAb
Ferritin heavy chain (B-12)	sc-376594	Santa Cruz	1:500	m mAb
Fibronectin	AB1954	Sigma-Aldrich	1:3000	r pAb
Vimentin	Ab92547	Abcam	1:3000	r mAb
GAPDH (6C5)	sc-32233	Santa Cruz	1:1000	m mAb
goat anti-rabbit IgG (H + L), HRP	#31460	Invitrogen	1:5000	g
goat anti-mouse IgG (H + L), HRP	#31430	Invitrogen	1:5000	r

^1^ Abbreviations are: g, goat; m, mouse; mAb, monoclonal antibody; pAb, polyclonal antibody; r, rabbit.

**Table 2 cells-14-00021-t002:** Collagen gene expression in Rat-1 cells as determined by next-generation sequencing.

Transcript Id ^1^	Gene Id	Gene	Gene Description	TPM	Remarks
ENSRNOT00000005311	ENSRNOG00000003897	*Col1a1*	collagen type I alpha 1 chain	1161.20482	type I collagens
ENSRNOT00000115180				26.056264	
ENSRNOT00000115096				8.904351	
ENSRNOT00000114204				1.004156	
ENSRNOT00000016423	ENSRNOG00000011292	*Col1a2*	collagen type I alpha 2 chain	1234.75059	
ENSRNOT00000087619				133.293852	
ENSRNOT00000004956	ENSRNOG00000003357	*Col3a1*	collagen type III alpha 1 chain	327.527538	type I collagen
ENSRNOT00000057386	ENSRNOG00000016281	*Col4a1*	collagen type IV alpha 1 chain	29.323839	type IV collagens
ENSRNOT00000114480				17.300902	
ENSRNOT00000057461	ENSRNOG00000023972	*Col4a2*	collagen type IV alpha 2 chain	42.735536	
ENSRNOT00000102469				1.011637	
ENSRNOT00000064478	ENSRNOG00000018951	*Col4a5*	collagen type IV alpha 5 chain	15.986744	
ENSRNOT00000108925				14.532791	
ENSRNOT00000012334	ENSRNOG00000008749	*Col5a1*	collagen type V alpha 1 chain	62.698572	type V collagens
ENSRNOT00000086352				30.084338	
ENSRNOT00000110258				41.388583	
ENSRNOT00000005073	ENSRNOG00000003736	*Col5a2*	collagen type V alpha 2 chain	198.382054	
ENSRNOT00000099326				131.062922	
ENSRNOT00000114323				12.814267	
ENSRNOT00000027897	ENSRNOG00000020525	*Col5a3*	collagen type V alpha 3 chain	39.529687	
ENSRNOT00000001679	ENSRNOG00000001249	*Col6a1*	collagen type VI alpha 1 chain	46.255988	type VI collagens
ENSRNOT00000001695	ENSRNOG00000001254	*Col6a2*	collagen type VI alpha 2 chain	49.057762	
ENSRNOT00000097191				18.272618	
ENSRNOT00000097191				18.272618	
ENSRNOT00000109457				5.469857	
ENSRNOT00000060838	ENSRNOG00000039668	*Col8a1*	collagen type VIII alpha 1 chain	21.738405	type VIII collagens
ENSRNOT00000113782				18.737532	
ENSRNOT00000014388	ENSRNOG00000010841	*Col8a2*	collagen type VIII alpha 2 chain	12.958648	
ENSRNOT00000068413	ENSRNOG00000023148	*Col11a1*	collagen type XI alpha 1 chain	2.05462	type XI collagen
ENSRNOT00000077071	ENSRNOG00000058470	*Col12a1*	collagen type XII alpha 1 chain	4.675117	type XII collagen
ENSRNOT00000067663	ENSRNOG00000031475	*Col16a1*	collagen type XVI alpha 1 chain	17.857773	type XVI collagen
ENSRNOT00000067663				17.857773	
E NSRNOG00000031475				17.317029	
ENSRNOT00000114962				10.859752	
ENSRNOT00000116489	ENSRNOG00000012110	*Col17a1*	collagen type XVII alpha 1 chain	1.855887	type XVII collagen
ENSRNOT00000095032	ENSRNOG00000001229	*Col18a1*	collagen type XVIII alpha 1 chain	199.064806	type XVIII collagen
ENSRNOT00000116507				1.801431	
ENSRNOT00000040270	ENSRNOG00000010326	*Col20a1*	collagen type XX alpha 1 chain	2.754775	type XX collagen
ENSRNOT00000010333	ENSRNOG00000007657	*Col27a1*	collagen type XXVII alpha 1 chain	5.073043	type XXVII collagen

^1^ Only collagen transcripts that were expressed higher than 1 Transcripts Per Million (TPM) are shown. Transcripts expressed at lower levels can be found in Appendix A.

**Table 3 cells-14-00021-t003:** Selected gene expression in Rat-1 cells underpinning their fibroblastic origin.

Transcript Id ^1^	Gene Id	*Gene*	Gene Description	TPM
ENSRNOT00000024430.5	ENSRNOG00000018087	*Vim*	Vimentin	9525.394084
ENSRNOT00000083468.1	ENSRNOG00000058039	*Acta2*	actin alpha 2, smooth muscle	23.318242
ENSRNOT00000015363	ENSRNOG00000011300	*Lama3*	laminin subunit alpha 3	5.52753
ENSRNOT00000092846				2.999081
ENSRNOT00000000737	ENSRNOG00000000599	*Lama4*	laminin subunit alpha 4	31.711221
ENSRNOT00000081226	ENSRNOG00000053691	*Lama5*	laminin subunit alpha 5	58.687106
ENSRNOT00000102342				1.055359
ENSRNOT00000095840	ENSRNOG00000005678	*Lamb1*	laminin subunit beta 1	100.330205
ENSRNOT00000008321				34.334559
ENSRNOT00000099819				2.716857
ENSRNOT00000072098	ENSRNOG00000047768	*Lamb2*	laminin subunit beta 2	86.84587
ENSRNOT00000083578				6.728922
ENSRNOT00000008440	ENSRNOG00000006025	*Lamb3*	laminin subunit beta 3	2.598395
ENSRNOT00000003605	ENSRNOG00000002680	*Lamc1*	laminin subunit gamma 1	102.954037
ENSRNOT00000036947	ENSRNOG00000002667	*Lamc2*	laminin subunit gamma 2	2.129861
ENSRNOT00000110905	ENSRNOG00000014288	*Fn1*	Fibronectin 1	453.772315
ENSRNOT00000057585				384.749075
ENSRNOT00000102568				0.051752
ENSRNOT00000013538	ENSRNOG00000009884	*Lgals1*	Galectin 1	8701.222759
ENSRNOT00000103260				9.71264
ENSRNOT00000115553				2.000297
ENSRNOT00000082304	ENSRNOG00000010645	*Lgals3*	Galectin 3	862.586359
ENSRNOT00000014216				26.612322
ENSRNOT00000082675				6.624262
ENSRNOT00000106924				5.688563
ENSRNOT00000027620	ENSRNOG00000020380	*Lgals7*	Galectin 7	1.196461
ENSRNOT00000044662	ENSRNOG00000018046	*Lgals8*	Galectin 8	51.377283
ENSRNOT00000107534	ENSRNOG00000018046	*Lgals8*	Galectin 8	5.089865
ENSRNOT00000017071	ENSRNOG00000012681	*Lgals9*	Galectin 9	1.300495
ENSRNOT00000017042				1.011874
ENSRNOT00000017486	ENSRNOG00000012840	*Sparc*	secreted protein acidic and cysteine rich	598.428609
ENSRNOT00000098312				374.064879
ENSRNOT00000115607				1.915055
ENSRNOT00000080598	ENSRNOG00000003772	*Csrp2*	cysteine and glycine-rich protein 2	10.21015
ENSRNOT00000067011				4.631306
ENSRNOT00000097951				1.276427
ENSRNOT00000028051	ENSRNOG00000020652	*Tgfb1*	transforming growth factor, beta 1	144.450709
ENSRNOT00000001775	ENSRNOG00000001312	*Pdgfa*	platelet derived growth factor subunit A	21.455048
ENSRNOT00000091476				5.50613
ENSRNOT00000042117				1.090084
ENSRNOT00000023196	ENSRNOG00000017197	*Pdgfb*	platelet derived growth factor subunit B	7.38854
ENSRNOT00000019501	ENSRNOG00000014350	*Ccn1*	cellular communication network factor 1	50.849188
ENSRNOT00000020528	ENSRNOG00000015036	*Ccn2*	cellular communication network factor 2	88.002219
ENSRNOT00000083468	ENSRNOG00000058039	*Acta2*	actin alpha 2, smooth muscle	23.318242
ENSRNOT00000105242	ENSRNOG00000034254	*Actb*	actin, beta	5508.934474
ENSRNOT00000042459				266.198453
*ENSRNOT00000050443*	*ENSRNOG00000018630*	*Gapdh*	glyceraldehyde-3-phosphate dehydrogenase	5800.944771
*ENSRNOT00000110793*				1732.710816
*ENSRNOT00000041328*				316.338683
*ENSRNOT00000114924*				77.734969

^1^ Only gene transcripts with expression levels higher than 1 Transcript Per Million (TPM) are shown. For comparison of transcript levels of the listed genes, the expressions of *Actb* and *Gapdh* (LOC108351137) are depicted. The complete mRNA expression profile of Rat-1 cells observed by NGS can be found in Appendix A.

**Table 4 cells-14-00021-t004:** Serum-induced circadian expressed genes according to [33].

Gene Id	Transcript Id	TPM	Gene Name ^1^	Gene Decription
ENSRNOG00000018413	ENSRNOT00000024932 ENSRNOT00000108999	13.03116 6.466574	*Per3*	period circadian regulator 3
ENSRNOG00000020254	ENSRNOT00000027506 ENSRNOT00000109112	8.795997 0.138503	*Per2*	period circadian regulator 2
ENSRNOG00000007387	ENSRNOT00000057136	8.631467	*Per1*	period circadian regulator 1
ENSRNOG00000009329	ENSRNOT00000012537 ENSRNOT00000117389	47.685616 1.310786	*Rev-Erba* ^2^	subfamily 1, group D, member 1
ENSRNOG00000021027	ENSRNOT00000028546 ENSRNOT00000091597	54.825901 3.809623	*Dbp*	D-box binding PAR bZIP transcription factor
ENSRNOG00000019383	ENSRNOT00000026258 ENSRNOT00000107707	38.963064 4.017308	*Tef*	TEF transcription factor, PAR bZIP family member
ENSRNOG00000008015	ENSRNOT00000010712	1.760769	*Fos*	Fos proto-oncogene, AP-1 transcription factor subunit

^1^ Abbreviations used: Dbp, D-box-binding Par bZIP transcription factor; Nr1d1, nuclear receptor subfamily 1, group D, member 1; Per1/2/3, period circadian regulator 1/2/3; TEF, thyrotrophic embryonic factor. ^2^ The official gene symbol approved by the HUGO Gene Nomenclature Committee is *Nr1d1* (for details see: https://www.genenames.org/data/gene-symbol-report/#!/symbol/NR1D1; accessed on 3 December 2024).

**Table 5 cells-14-00021-t005:** Losses and gains of chromosomal regions in Rat-1 cells according to mFISH and inverted DAPI-banding analyses. For each region, the cytogenetic span and approximate molecular span in the rat genome are provided, as well as the approximate cytogenetic span when projected onto the human genome. In case breakpoints were not exactly determined, they were transformed into the most likely regions of CNAs as highlighted by a => Number of gains or losses are indicated in curly brackets.

Cytogenetic Span	Molecular Span [RGSC 5.0/rn5]	Cytogenetic Span Based on Translation Into [GRCh37/hg19]
Gain		
2q?2->2q?2 {+4} => 2q22->2q25	80406983–139816749	5p15.31p14.1 3q26.2q26.33 8q21.11q21.3 5p13.3p13.2 8q12.3q13.1 3q24q25.1
6q11->6qter {+1}	1–156897508	14q12q32.33 2pterp16.3 7p21.2p15.3 7q22.3q31.1
7pter->7qter {+1}	1–143501887	12q12q23.3 8q22.1q24.3 22q12.3q13.33
10q?3-10qter {+1} => 10q32.1->10qter	89211697–112200500	17q21.31qter
12pter->q1?2 {+1} => 12pter->q12	1–31247709	13q12.13q13.2 7pterp22.1 7q11.21q11.22 7q22.1q22.1 7q21.2q22.1 7q11.23q11.23
19pter->19p11 {+2}	1–27543415	16q12.2q22.1 22q12.3q12.3
19p11->19qter {+1}	27543416–72914587	16q22.1qter 16q11.2q12.2 10p11.22p11.21 19p13.2p13.12 1q24.13q24.3
Loss		
2q34->2qter {−1}	192680326–285068071	1p31.3q23.1 4q22.3q26 4q28.1q32.2
3q?22->3qter {−1} => 3q22->3qter	58164792–183740530	20p13qter 11p14.2q12.1 2q24.3q32.1 15q14q21.2
4pter->4qter {−1}	1–248343840	7q11.23q34 3p14.1p12.3 12p13.33p11.21 7p15.3p14.3 3p26.3p25.2 2p13.3p11.2 7q34q36.1 7p22.1p21.3
8pter->8qter {−1}	1–132457389	11q22.3qter 3p24.1q24 15q21.2q24.3 6q13q14.3 11q14.3q22.3 3p22.2p21.31 7p14.3p14.2

**Table 6 cells-14-00021-t006:** Losses and gains of chromosomal regions in Rat-1 cells translated into the human genome compared with gains and losses of these regions in epithelioid sarcoma and liposarcoma ^1^.

Human Chromosomal Regions	Rat-1	Epithelioid Sarcoma [23]	Liposarcoma [24]
1p31.3q23.1	loss	**loss**	no CAN
1q24.13q24.3	gain	**gain**	**gain**
2pterp16.3	gain	**gain**	no CNA
2p13.3p11.2	loss	gain	no CNA
2q24.3q32.1	loss	no CNA	**loss** and gain
3p26.3p25.2	loss	no CNA	no CNA
3p24.1q24	loss	no CNA	no CNA
3p22.2p21.31	loss	**loss**	no CNA
3p14.1p12.3	loss	**loss**	no CNA
3q24q25.1	gain	no CNA	no CNA
3q26.2q26.33	gain	no CNA	no CNA
4q22.3q26	loss	**loss**	no CNA
4q28.1q32.2	loss	**loss**	no CNA
5p15.31p14.1	gain	no CNA	**gain**
5p13.3p13.2	gain	no CNA	**gain**
6q13q14.3	loss	**loss**	**loss**
7pterp22.1	gain	**gain**	**gain**
7p22.1p21.3	loss	gain	no CNA
7p21.2p15.3	gain	**gain**	no CNA
7p15.3p14.2	loss	gain	no CNA
7q11.21q11.23	gain	**gain**	no CNA
7q11.23q21.2	loss	gain	no CNA
7q21.2q22.1	gain	**gain**	no CNA
7q22.3q31.1	gain	**gain**	no CNA
7q31.1q36.1	loss	gain	no CNA
8q12.3q13.1	gain	**gain**	**gain**
8q21.11q24.3	gain	**gain**	**gain**
10p11.22p11.21	gain	no CNA	no CNA
11p14.2q12.1	loss	no CNA	no CNA
11q14.3qter	loss	no CNA	**loss**
12p13.33p11.21	loss	**loss**	no CNA
12q12q23.3	gain	no CNA	**gain**
13q12.13q13.2	gain	loss	loss and **gain**
14q12q32.33	gain	no CNA	**gain**
15q14q24.3	loss	gain	**loss**
16q11.2qter	gain	**gain**	loss
17q21.31qter	gain	no CNA	**gain**
19p13.2p13.12	gain	no CNA	**gain**
20p13qter	loss	gain	gain
22q12.3q13.33	gain	no CNA	loss
overall		17/40	15/40

^1^ Regions with common gain or loss in human sarcomas are highlighted in bold.

**Table 7 cells-14-00021-t007:** STR-based DNA profiling of Rat 1 cells using the 31 species specific STR markers.

SN	Marker Name ^1^	Location on Chromosome	Allele Sizes (bp) in CFSC-2G	Allele Sizes (bp) in HSC-T6	Allele Sizes (bp) in PAV-1	Allele Sizes (bp) in Rat-1
1	73	1	194, 203	194	194	211, 213
2	8	2	236	234	234, 238	232, 236
3	2	2	126	127	128	129
4	4	3	268, 270	238	236, 238	250, 252
5	3	3	160, 182	160, 162	162	178, 182
6	26	4	150	166	154	162
7	19	4	180	175	179	176, 178
8	81	5	130, 134	130, 132	130	128
9	34	6	184, 189	188	182, 187	184, 189
10	30	7	188, 192	186, 192	192	186
11	24	8	260	249, 253	254, 259	247, 249
12	59	9	145	143, 146, 180	145, 148	176, 178
13	62	9	166	177	166	154
14	1	10	105	96	96, 105	96
15	55	10	210, 214	210, 218	210, 218	203, 205
16	36	11	222	234	222	228
17	67	11	154, 156	165	165	165, 167
18	13	12	121	121, 135	121	121
19	35	13	197	197, 203	203	203
20	42	13	125	127	144, 156	154, 156
21	70	14	158, 175	175, 179	158, 175	158
22	61	15	128	128	128	110
23	79	15	172, 180	172	172	172
24	90	16	159, 161	174	175	159, 161
25	69	16	138	139	136, 139	148
26	78	17	136, 151	147, 151	147, 149	136, 140
27	15	18	232	232	232	238
28	16	18	251, 260	247, 251	251	247, 251
29	75	19	144	144, 184	144, 184	144
30	96	20	210	210, 212	210	208, 210
31	91	20	221	205, 211	211, 225	219, 221

^1^ Testing was conducted using the CellCheck^TM^ Rat Panel (IDEXX BioAnalytics, Columbia, MO, USA).

## Data Availability

The original contributions presented in the study are included in the article and Appendix A; further inquiries can be directed to the corresponding author.

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
