# Peer review of "Genetic Characteristics of the Rat Fibroblast Cell Line Rat-1"

_cells, 2024, doi:10.3390/cells14010021_

Round 1

Reviewer 1 Report

Comments and Suggestions for Authors

"Genetic Characteristics of the Rat Fibroblast Cell Line Rat-1" is an interesting work that increases our knowledge of Rat-1 cell line. In this manuscript, Liehr et al performed thorough genomic analyses (e.g., karyotype analysis, STR, and RNA-seq), morphological and biochemical experiments on Rat-1 cell line. It is known that the expression of circadian related genes in Rat-1 cells is affected by serum shock (Cell. 1998 Jun 12;93(6):929-37. doi: 10.1016/s0092-8674(00)81199-x.). As the authors did RNA-seq, I think the dynamic expression pattern of Rat-1 under serum shock would have been more interesting than identifying the expression of genes related to its fibroblast origin.

Author Response

Dear Reviewer 1,

many thanks for your excellent suggestions. Please find our comments to your comments in the attached pdf-file.

Regards

Ralf Weiskirchen

Reviewer 2 Report

Comments and Suggestions for Authors

The authors propose a manuscript that describes the genetic characteristics of the Rat-1 cell line.

The overall goal of the manuscript is dual:

a.       Represent a source that provides a description and characterization of the cell line, with a more detailed analysis of chromosomal organization, transcriptional profile, STR profiling and a general description of cellular organization and sub-cellular structures;

b.       Provide insights about the complex chromosomal re-arrangements observed in a cell line derived from a rat embryo and that underwent spontaneous immortalization, leading to the conclusion that these cells should not be used as normal fibroblasts but as a reliable model system for human sarcoma (lines 467-469).

I found that the part related to the complex chromosomal rearrangements and STR profiling was quite well described and of interest. However, there are some key aspects that require revision.

Related to the point a., the proposed manuscript successfully describes this cell line. However, most of the descriptions are short of statistical analysis. I could not find the exact number of times each experiment was performed. This is relevant in particular for quantitative experiments, like NGS sequencing that led to analysis about known fibroblast-specific genes and their expression. The authors indicate that the described “genes are expressed in Rat-1 cells, supporting the idea that Rat-1 cells are of fibroblastic origin” (lines 264-265). Although I would agree with the conclusion that these genes are widely known to be expressed at reasonable level by fibroblasts, no comparison with other cells (for example, non-fibroblast line) is performed and the threshold used (>1 TPM) looks to be with a very low stringency, leading to the question of whether some of the proteins associated with the genes in Table 3 would be really detectable or not.

As for the point b., the authors comment that “the genetic stability and potential variations of Rat-1 cells over extended culture periods are poorly understood” (lines 18-19). Moreover, authors suggest that the chromosomal instability present in this cell line could impact research and caution should be used (lines 458-464). Finally, the authors suggest that these cells should not be used as normal fibroblasts but as a reliable model system for human sarcoma (lines 467-469).

However, the authors did not perform a temporal analysis of these complex to understand how much the chromosomal instability (CIN) affects the stability of the cell line. The data suggest that this cell line was already aneuploid when received from RIKEN institute, which is compatible with the fact that this model system underwent spontaneous immortalization and was established more than 50 years ago. I agree with the authors that it is important to report this finding, but without a temporal analysis authors should not make claims about stability of the model, as it is unclear if this was an event happened at the beginning of the selection or not.

Moreover, the authors have nor performed a comparative analysis of different Rat-1 cells obtained from different sources which would have helped to extend the relevance of their findings and implications for the scientific community, therefore I would suggest that the authors include a comment about this in the discussion section.

Finally, if the author believe that these data suggest that Rat-1 cell line should not be used as a normal fibroblast line, it becomes difficult to believe that these cells could be used as a model of human sarcoma. Indeed, they have not been originally isolated from a tumour and their transformation process is unknown.

Minor comments:

-          In Figure 1, the scale bar should describe length in micro-meters;

-          The F-actin cytoskeleton in Figure 4 is under the paragraph 3.5.2 named “Western Blot analysis”. Given that Figure 4 is not a Western blot, I would suggest to correct the paragraph name with something more appropriate.

Author Response

Dear Reviewer 2,

many thanks for your excellent suggestions. Please find our comments to your comments in the attached pdf-file.

Regards

Ralf Weiskirchen

Round 2

Reviewer 2 Report

Comments and Suggestions for Authors

The authors addressed my points.